# Reward Constrained Policy Optimization

Chen Tessler[1], Daniel J. Mankowitz[2], and Shie Mannor[1]

[1] *Technion Israel Institute of Technology, Haifa, Israel*
[2] *DeepMind, London, England*
 *chen.tessler@campus.technion.ac.il, dmankowitz@google.com, shie@ee.technion.ac.il*

## Abstract

Solving tasks in Reinforcement Learning is no easy feat. As the goal of the agent is to maximize the accumulated reward, it often learns to exploit loopholes and misspecifications in the reward signal resulting in unwanted behavior. While constraints may solve this issue, there is no closed form solution for general constraints. In this work, we present a novel multi-timescale approach for constrained policy optimization, called 'Reward Constrained Policy Optimization' (RCPO), which uses an alternative penalty signal to guide the policy towards a constraint satisfying one. We prove the convergence of our approach and provide empirical evidence of its ability to train constraint satisfying policies.

## 1 Introduction

Applying Reinforcement Learning (RL) is generally a hard problem. At each state, the agent performs an action which produces a reward. The goal is to maximize the accumulated reward, hence the reward signal implicitly defines the behavior of the agent. While in computer games (e.g. Bellemare et al. (2013)) there exists a pre-defined reward signal, it is not such in many real applications.

An example is the Mujoco domain (Todorov et al., 2012), in which the goal is to learn to control robotic agents in tasks such as: standing up, walking, navigation and more. Considering the Humanoid domain, the agent is a 3 dimensional humanoid and the task is to walk forward as far as possible (without falling down) within a fixed amount of time. Naturally, a reward is provided based on the forward velocity in order to encourage a larger distance; however, additional reward signals are provided in order to guide the agent, for instance a bonus for staying alive, a penalty for energy usage and a penalty based on the force of impact between the feet and the floor (which should encourage less erratic behavior). Each signal is multiplied by it's own coefficient, which controls the emphasis placed on it.

This approach is a multi-objective problem (Mannor and Shimkin, 2004); in which for each set of penalty coefficients, there exists a different, optimal solution, also known as Pareto optimality (Van Moffaert and Nowé, 2014). In practice, the exact coefficient is selected through a time consuming and a computationally intensive process of *hyper-parameter* tuning. As our experiments show, the coefficient is not shared across domains, a coefficient which leads to a satisfying behavior on one domain may lead to catastrophic failure on the other (issues also seen in Leike et al. (2017) and Mania et al. (2018)). Constraints are a natural and consistent approach, an approach which ensures a satisfying behavior without the need for manually selecting the penalty coefficients.

In constrained optimization, the task is to maximize a target function $f(x)$ while satisfying an inequality constraint $g(x) \leq \alpha$. While constraints are a promising solution to ensuring a satisfying behavior, existing methods are limited in the type of constraints they are able to handle and the algorithms that they may support - they require a parametrization of the policy (policy gradient methods) and propagation of the constraint violation signal over the entire trajectory (e.g. Prashanth and Ghavamzadeh (2016)). This poses an issue, as Q-learning algorithms such as DQN (Mnih et al., 2015) do not learn a parametrization of the policy, and common Actor-Critic methods (e.g. (Schulman et al., 2015a; Mnih et al., 2016;

Table 1: Comparison between various approaches.

| | Handles discounted sum constraints | Handles mean value[1] constraints | Requires no prior knowledge | Reward agnostic |
|---|---|---|---|---|
| RCPO (this paper) | ✓ | ✓[2] | ✓ | ✓ |
| Dalal et al. (2018) | ✗ | ✗ | ✗ | ✓ |
| Achiam et al. (2017) | ✓ | ✗ | ✓ | ✓ |
| Reward shaping | ✓ | ✓ | ✓ | ✗ |
| Schulman et al. (2017)[3] | ✗ | ✗ | ✗ | ✗ |

Schulman et al., 2017)) build the reward-to-go based on an N-step sample and a bootstrap update from the critic.

In this paper, we propose the 'Reward Constrained Policy Optimization' (RCPO) algorithm. RCPO incorporates the constraint as a penalty signal into the reward function. This penalty signal guides the policy towards a constraint satisfying solution. We prove that RCPO converges almost surely, under mild assumptions, to a constraint satisfying solution (Theorem 2). In addition; we show, empirically on a toy domain and six robotics domains, that RCPO results in a constraint satisfying solution while demonstrating faster convergence and improved stability (compared to the standard constraint optimization methods).

**Related work:** Constrained Markov Decision Processes (Altman, 1999) are an active field of research. CMDP applications cover a vast number of topics, such as: electric grids (Koutsopoulos and Tassiulas, 2011), networking (Hou and Zhao, 2017), robotics (Chow et al., 2015; Gu et al., 2017; Achiam et al., 2017; Dalal et al., 2018) and finance (Krokhmal et al., 2002; Tamar et al., 2012).

The main approaches to solving such problems are (i) Lagrange multipliers (Borkar, 2005; Bhatnagar and Lakshmanan, 2012), (ii) Trust Region (Achiam et al., 2017), (iii) integrating prior knowledge (Dalal et al., 2018) and (iv) manual selection of the penalty coefficient (Tamar and Mannor, 2013; Levine and Koltun, 2013; Peng et al., 2018).

**Novelty:** The novelty of our work lies in the ability to tackle (1) general constraints (both discounted sum and mean value constraints), not only constraints which satisfy the recursive Bellman equation (i.e, discounted sum constraints) as in previous work. The algorithm is (2) reward agnostic. That is, invariant to scaling of the underlying reward signal, and (3) does not require the use of prior knowledge. A comparison with the different approaches is provided in Table 1.

## 2 Preliminaries

### 2.1 Markov Decision Process (MDP)

A Markov Decision Processes $\mathcal{M}$ is defined by the tuple $(S, A, R, P, \mu, \gamma)$ (Sutton and Barto, 1998). Where $S$ is the set of states, $A$ the available actions, $R : S \times A \times S \mapsto \mathbb{R}$ is the reward function, $P : S \times A \times S \mapsto [0, 1]$ is the transition matrix, where $P(s'|s, a)$ is the probability of transitioning from state $s$ to $s'$ assuming action $a$ was taken, $\mu : S \mapsto [0, 1]$ is the initial state distribution and $\gamma \in [0, 1)$ is the discount factor for future rewards. A policy $\pi : S \mapsto \Delta_A$ is a probability distribution over actions and $\pi(a|s)$ denotes the probability of taking action $a$ at state $s$. For each state $s$, the value of following policy $\pi$ is denoted by:

$$V_R^\pi(s) = \mathbb{E}^\pi[\sum_t \gamma^t r(s_t, a_t)|s_0 = s] \ .$$

---

[1] A mean valued constraint takes the form of $\mathbb{E}[\frac{1}{T} \sum_{t=0}^{T-1} c_t] \leq \alpha$, as seen in Section 5.2.

[2] Under the appropriate assumptions.

[3] Algorithms such as PPO are not intended to consider or satisfy constraints.

An important property of the value function is that it solves the recursive Bellman equation:

$$V_R^\pi(s) = \mathbb{E}^\pi[r(s,a) + \gamma V_R^\pi(s')|s] \ .$$

The goal is then to maximize the expectation of the reward-to-go, given the initial state distribution $\mu$:

$$\max_{\pi \in \Pi} J_R^\pi \ , \text{ where } \quad J_R^\pi = \mathbb{E}_{s \sim \mu}^\pi[\sum_{t=0}^\infty \gamma^t r_t] = \sum_{s \in S} \mu(s) V_R^\pi(s) \ . \tag{1}$$

## 2.2 CONSTRAINED MDPS

A Constrained Markov Decision Process (CMDP) extends the MDP framework by introducing a penalty $c(s,a)$, a constraint $C(s_t) = F(c(s_t, a_t), ..., c(s_N, a_N))$ and a threshold $\alpha \in [0, 1]$. A constraint may be a discounted sum (similar to the reward-to-go), the average sum and more (see Altman (1999) for additional examples). Throughout the paper we will refer to the collection of these constraints as **general constraints**.

We denote the expectation over the constraint by:

$$J_C^\pi = \mathbb{E}_{s \sim \mu}^\pi[C(s)] \ . \tag{2}$$

The problem thus becomes:

$$\max_{\pi \in \Pi} J_R^\pi \ , \ \texttt{s.t.} \quad J_C^\pi \leq \alpha \ . \tag{3}$$

## 2.3 PARAMETRIZED POLICIES

In this work we consider parametrized policies, such as neural networks. The parameters of the policy are denoted by $\theta$ and a parametrized policy as $\pi_\theta$. We make the following assumptions in order to ensure convergence to a constraint satisfying policy:

**Assumption 1.** *The value $V_R^\pi(s)$ is bounded for all policies $\pi \in \Pi$.*

**Assumption 2.** *Every local minima of $J_C^{\pi_\theta}$ is a feasible solution.*

Assumption 2 is the minimal requirement in order to ensure convergence, given a general constraint, of a gradient algorithm to a feasible solution. Stricter assumptions, such as convexity, may ensure convergence to the optimal solution; however, in practice constraints are non-convex and such assumptions do not hold.

## 3 CONSTRAINED POLICY OPTIMIZATION

Constrained MDP's are often solved using the Lagrange relaxation technique (Bertesekas, 1999). In Lagrange relaxation, the CMDP is converted into an equivalent unconstrained problem. In addition to the objective, a penalty term is added for infeasibility, thus making infeasible solutions sub-optimal. Given a CMDP (3), the unconstrained problem is

$$\min_{\lambda \geq 0} \max_\theta L(\lambda, \theta) = \min_{\lambda \geq 0} \max_\theta [J_R^{\pi_\theta} - \lambda \cdot (J_C^{\pi_\theta} - \alpha)] \ , \tag{4}$$

where $L$ is the Lagrangian and $\lambda \geq 0$ is the Lagrange multiplier (a penalty coefficient). Notice, as $\lambda$ increases, the solution to (4) converges to that of (3). This suggests a two-timescale approach: on the faster timescale, $\theta$ is found by solving (4), while on the slower timescale, $\lambda$ is increased until the constraint is satisfied. The goal is to find a saddle point $(\theta^*(\lambda^*), \lambda^*)$ of (4), which is a feasible solution.

**Definition 1.** *A feasible solution of the CMDP is a solution which satisfies $J_C^\pi \leq \alpha$.*

### 3.1 Estimating the gradient

We assume there isn't access to the MDP itself, but rather samples are obtained via simulation. The simulation based algorithm for the constrained optimization problem (3) is:

$$\lambda_{k+1} = \Gamma_\lambda[\lambda_k - \eta_1(k)\nabla_\lambda L(\lambda_k, \theta_k)] \ , \tag{5}$$

$$\theta_{k+1} = \Gamma_\theta[\theta_k + \eta_2(k)\nabla_\theta L(\lambda_k, \theta_k)] \ , \tag{6}$$

where $\Gamma_\theta$ is a projection operator, which keeps the iterate $\theta_k$ stable by projecting onto a compact and convex set. $\Gamma_\lambda$ projects $\lambda$ into the range $[0, \lambda_{\max}{}^4]$. $\nabla_\theta L$ and $\nabla_\lambda L$ are derived from (4), where the formulation for $\nabla_\theta L$ is derivied using the log-likelihood trick (Williams, 1992):

$$\nabla_\theta L(\lambda, \theta) = \nabla_\theta \mathbb{E}^{\pi_\theta}_{s\sim\mu} [\log \pi(s, a; \theta) [R(s) - \lambda \cdot C(s)]] \ , \tag{7}$$

$$\nabla_\lambda L(\lambda, \theta) = -(\mathbb{E}^{\pi_\theta}_{s\sim\mu}[C(s)] - \alpha) \ , \tag{8}$$

$\eta_1(k), \eta_2(k)$ are step-sizes which ensure that the policy update is performed on a faster timescale than that of the penalty coefficient $\lambda$.

**Assumption 3.**

$$\sum_{k=0}^\infty \eta_1(k) = \sum_{k=0}^\infty \eta_2(k) = \infty, \ \ \sum_{k=0}^\infty \left(\eta_1(k)^2 + \eta_2(k)^2\right) < \infty \ \ and \ \ \frac{\eta_1(k)}{\eta_2(k)} \to 0 \ .$$

**Theorem 1.** *Under Assumption 3, as well as the standard stability assumption for the iterates and bounded noise (Borkar et al., 2008), the iterates $(\theta_n, \lambda_n)$ converge to a fixed point (a local minima) almost surely.*

**Lemma 1.** *Under assumptions 1 and 2, the fixed point of Theorem 1 is a feasible solution.*

The proof to Theorem 1 is provided in Appendix C and to Lemma 1 in Appendix D.

## 4 Reward Constrained Policy Optimization

### 4.1 Actor Critic Requirements

Recently there has been a rise in the use of Actor-Critic based approaches, for example: A3C (Mnih et al., 2016), TRPO (Schulman et al., 2015a) and PPO (Schulman et al., 2017). The actor learns a policy $\pi$, whereas the critic learns the value (using temporal-difference learning - the recursive Bellman equation). While the original use of the critic was for variance reduction, it also enables training using a finite number of samples (as opposed to Monte-Carlo sampling).

Our goal is to tackle general constraints (Section 2.2), as such, they are not ensured to satisfy the recursive property required to train a critic.

### 4.2 Penalized reward functions

We overcome this issue by training the actor (and critic) using an alternative, guiding, penalty - the discounted penalty. The appropriate assumptions under which the process converges to a feasible solution are provided in Theorem 2. It is important to note that; in order to ensure constraint satisfaction, $\lambda$ is still optimized using Monte-Carlo sampling on the original constraint (8).

**Definition 2.** *The value of the discounted (guiding) penalty is defined as:*

$$V^\pi_{C_\gamma}(s) \triangleq \mathbb{E}^\pi \left[\sum_{t=0}^\infty \gamma^t c(s_t, a_t)|s_0 = s\right] \ . \tag{9}$$

---

[4]When Assumption 2 holds, $\lambda_{\max}$ can be set to $\infty$.

**Definition 3.** *The penalized reward functions are defined as:*

$$\hat{r}(\lambda, s, a) \triangleq r(s, a) - \lambda c(s, a) \ , \tag{10}$$

$$\hat{V}^\pi(\lambda, s) \triangleq \mathbb{E}^\pi \left[ \sum_{t=0}^\infty \gamma^t \hat{r}(\lambda, s_t, a_t) | s_0 = s \right]$$

$$= \mathbb{E}^\pi \left[ \sum_{t=0}^\infty \gamma^t \left( r(s_t, a_t) - \lambda c(s_t, a_t) \right) | s_0 = s \right] = V_R^\pi(s) - \lambda V_{C_\gamma}^\pi(s) \ . \tag{11}$$

As opposed to (4), for a fixed $\pi$ and $\lambda$, the penalized value (11) can be estimated using TD-learning critic. We denote a three-timescale (Constrained Actor Critic) process, in which the actor and critic are updated following (11) and $\lambda$ is updated following (5), as the 'Reward Constrained Policy Optimization' (RCPO) algorithm. Algorithm 1 illustrates such a procedure and a full RCPO Advantage-Actor-Critic algorithm is provided in Appendix A.

---

**Algorithm 1** Template for an RCPO implementation

---

1: **Input:** penalty $c(\cdot)$, constraint $C(\cdot)$, threshold $\alpha$, learning rates $\eta_1(k) < \eta_2(k) < \eta_3(k)$
2: Initialize actor parameters $\theta = \theta_0$, critic parameters $v = v_0$, Lagrange multipliers and $\lambda = 0$
3: **for** $k = 0, 1, ...$ **do**
4:     Initialize state $s_0 \sim \mu$
5:     **for** $t = 0, 1, ..., T - 1$ **do**
6:         Sample action $a_t \sim \pi$, observe next state $s_{t+1}$, reward $r_t$ and penalties $c_t$
7:         $\hat{R}_t = r_t - \lambda_k c_t + \gamma \hat{V}(\lambda, s_t; v_k)$                                                    ▷ Equation 10
8:         **Critic update:** $v_{k+1} \leftarrow v_k - \eta_3(k) \left[ \partial(\hat{R}_t - \hat{V}(\lambda, s_t; v_k))^2 / \partial v_k \right]$     ▷ Equation 11
9:         **Actor update:** $\theta_{k+1} \leftarrow \Gamma_\theta \left[ \theta_k + \eta_2(k) \nabla_\theta \hat{V}(\lambda, s) \right]$     ▷ Equation 6
10:        **Lagrange multiplier update:** $\lambda_{k+1} \leftarrow \Gamma_\lambda \left[ \lambda_k + \eta_1(k) \left( J_C^{\pi_\theta} - \alpha \right) \right]$     ▷ Equation 8
11: **return** policy parameters $\theta$

---

**Theorem 2.** *Denote by $\Theta = \{\theta : J_C^{\pi_\theta} \leq \alpha\}$ the set of feasible solutions and the set of local-minimas of $J_{C_\gamma}^{\pi_\theta}$ as $\Theta_\gamma$. Assuming that $\Theta_\gamma \subseteq \Theta$ then the 'Reward Constrained Policy Optimization' (RCPO) algorithm converges almost surely to a fixed point $(\theta^*(\lambda^*, v^*), v^*(\lambda^*), \lambda^*)$ which is a feasible solution (e.g. $\theta^* \in \Theta$).*

The proof to Theorem 2 is provided in Appendix E.

The assumption in Theorem 2 demands a specific correlation between the guiding penalty signal $C_\gamma$ and the constraint $C$. Consider a robot with an average torque constraint. A policy which uses 0 torque at each time-step is a feasible solution and in turn is a local minimum of both $J_C$ and $J_{C_\gamma}$. If such a policy is reachable from any $\theta$ (via gradient descent), this is enough in order to provide a theoretical guarantee such that $J_{C_\gamma}$ may be used as a guiding signal in order to converge to a fixed-point, which is a feasible solution.

## 5 EXPERIMENTS

We test the RCPO algorithm in various domains: a grid-world, and 6 tasks in the Mujoco simulator (Todorov et al., 2012). The grid-world serves as an experiment to show the benefits of RCPO over the standard Primal-Dual approach (solving (4) using Monte-Carlo simulations), whereas in the Mujoco domains we compare RCPO to reward shaping, a simpler (yet common) approach, and show the benefits of an adaptive approach to defining the cost value.

While we consider mean value constraints (robotics experiments) and probabilistic constraints (i.e., Mars rover), discounted sum constraints can be immediately incorporated into

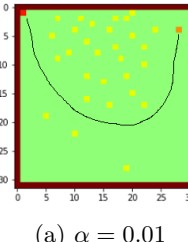 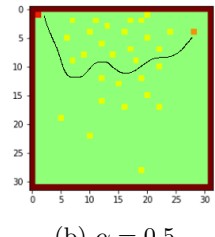

(a) $\alpha = 0.01$           (b) $\alpha = 0.5$

Figure 1: Mars Rover domain and policy illustration. As $\alpha$ decreases, the agent is required to learn a safer policy.

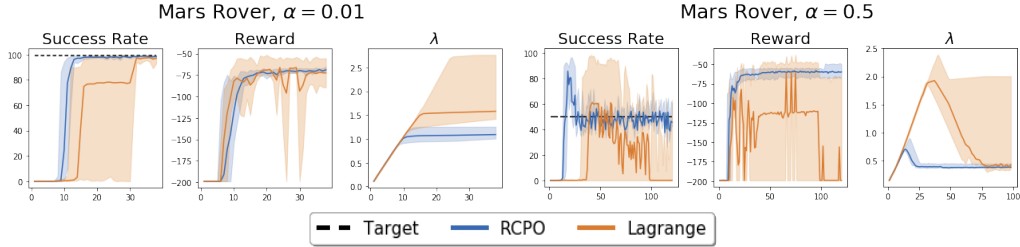

Figure 2: RCPO vs Lagrange comparison. The reward is $(-)$ the average number of steps it takes to reach the goal. Results are considered valid if and only if they are at or below the threshold.

our setup. We compare our approach with relevant baselines that can support these constraints. Discounted sum approaches such as Achiam et al. (2017) and per-state constraints such as Dalal et al. (2018) are unsuitable for comparison given the considered constraints. See Table 1 for more details.

For clarity, we provide exact details in Appendix B (architecture and simulation specifics).

## 5.1 MARS ROVER

### 5.1.1 DOMAIN DESCRIPTION

The rover (red square) starts at the top left, a safe region of the grid, and is required to travel to the goal (orange square) which is located in the top right corner. The transition function is stochastic, the rover will move in the selected direction with probability $1 - \delta$ and randomly otherwise. On each step, the agent receives a small negative reward $r_{\text{step}}$ and upon reaching the goal state a reward $r_{\text{goal}}$. Crashing into a rock (yellow) causes the episode to terminate and provides a negative reward $-\lambda$. The domain is inspired by the Mars Rover domain presented in Chow et al. (2015). It is important to note that the domain is built such that a shorter path induces higher risk (more rocks along the path). Given a minimal failure threshold ($\alpha \in (0, 1)$), the task is to find $\lambda$, such that when solving for parameters $\delta, r_{\text{step}}, r_{\text{goal}}$ and $\lambda$, the policy will induce a path with $\mathbb{P}^{\pi_\theta}_\mu(\text{failure}) \leq \alpha$; e.g., find the shortest path while ensuring that the probability of failure is less or equal to $\alpha$.

### 5.1.2 EXPERIMENT DESCRIPTION

As this domain is characterized by a discrete action space, we solve it using the A2C algorithm (a synchronous version of A3C (Mnih et al., 2016)). We compare RCPO, using the discounted penalty $C_\gamma$, with direct optimization of the Lagrange dual form (4).

### 5.1.3 EXPERIMENT ANALYSIS

Figure 1 illustrates the domain and the policies the agent has learned based on different safety requirements. Learning curves are provided in Figure 2. The experiments show that,

Table 2: Comparison between RCPO and reward shaping with a torque constraint $< 25\%$.

|  | Swimmer-v2 | | Walker2d-v2 | | Hopper-v2 | |
|---|---|---|---|---|---|---|
|  | Torque | Reward | Torque | Reward | Torque | Reward |
| $\lambda = 0$ | 30.4% | 94.4 | **24.6%** | **3364.1** | 31.5% | 2610.7 |
| $\lambda = 0.00001$ | 37.4% | 65.1 | 28.4% | 3198.9 | 31.4% | 1768.2 |
| $\lambda = 0.1$ | 32.8% | 16.5 | 13.6% | 823.5 | **15.7%** | **865.9**[5] |
| $\lambda = 100$ | 2.4% | 11.7 | 17.8% | 266.1 | 14.3% | 329.4 |
| **RCPO (ours)** | **24%** | **72.7** | 25.2% | 591.6 | **26%** | **1138.5**[5] |

|  | Humanoid-v2 | | HalfCheetah-v2 | | Ant-v2 | |
|---|---|---|---|---|---|---|
|  | Torque | Reward | Torque | Reward | Torque | Reward |
| $\lambda = 0$ | 28.6% | 617.1 | 37.8% | 2989.5 | 36.7% | 1313.1 |
| $\lambda = 0.00001$ | 28.1% | 617.1 | 40.8% | 2462.3 | 35.9% | 1233.5 |
| $\lambda = 0.1$ | 28.5% | 1151.8 | 13.87% | -0.4 | 16.6% | 1012.2 |
| $\lambda = 100$ | 30.5% | 119.4 | 13.9% | -2.4 | 16.7% | 957.2 |
| **RCPO (ours)** | **24.3%** | **606.1** | **26.7%** | **1547.1** | **15.2%** | **1031.5** |

for both scenarios $\alpha = 0.01$ and $\alpha = 0.5$, RCPO is characterized by faster convergence (improved sample efficiency) and lower variance (a stabler learning regime).

## 5.2 ROBOTICS

### 5.2.1 DOMAIN DESCRIPTION

Todorov et al. (2012); Brockman et al. (2016) and OpenAI (2017) provide interfaces for training agents in complex control problems. These tasks attempt to imitate scenarios encountered by robots in real life, tasks such as teaching a humanoid robot to stand up, walk, and more. The robot is composed of $n$ joints; the state $S \in \mathbb{R}^{n \times 5}$ is composed of the coordinates $(x, y, z)$ and angular velocity $(\omega_\theta, \omega_\phi)$ of each joint. At each step the agent selects the amount of torque to apply to each joint. We chose to use PPO (Schulman et al., 2017) in order to cope with the continuous action space.

### 5.2.2 EXPERIMENT DESCRIPTION

In the following experiments; the aim is to prolong the motor life of the various robots, while still enabling the robot to perform the task at hand. To do so, the robot motors need to be constrained from using high torque values. This is accomplished by defining the constraint $C$ as the average torque the agent has applied to each motor, and the per-state penalty $c(s, a)$ becomes the amount of torque the agent decided to apply at each time step. We compare RCPO to the reward shaping approach, in which the different values of $\lambda$ are selected apriori and remain constant.

### 5.2.3 EXPERIMENT ANALYSIS

Learning curves are provided in Figure 3 and the final values in Table 2. It is important to note that by preventing the agent from using high torque levels (limit the space of admissible policies), the agent may only be able to achieve a sub-optimal policy. RCPO aims to find the best performing policy given the constraints; that is, the policy that achieves maximal value while at the same time satisfying the constraints. Our experiments show that:

1. In all domains, RCPO finds a feasible (or near feasible) solution, and, besides the Walker2d-v2 domain, exhibits superior performance when compared to the relevant reward shaping variants (constant $\lambda$ values resulting in constraint satisfaction).

---

[5]In the Hopper-v2 domain, it is not clear which is the superior method RCPO or the $\lambda = 0.1$ reward shaping variant.

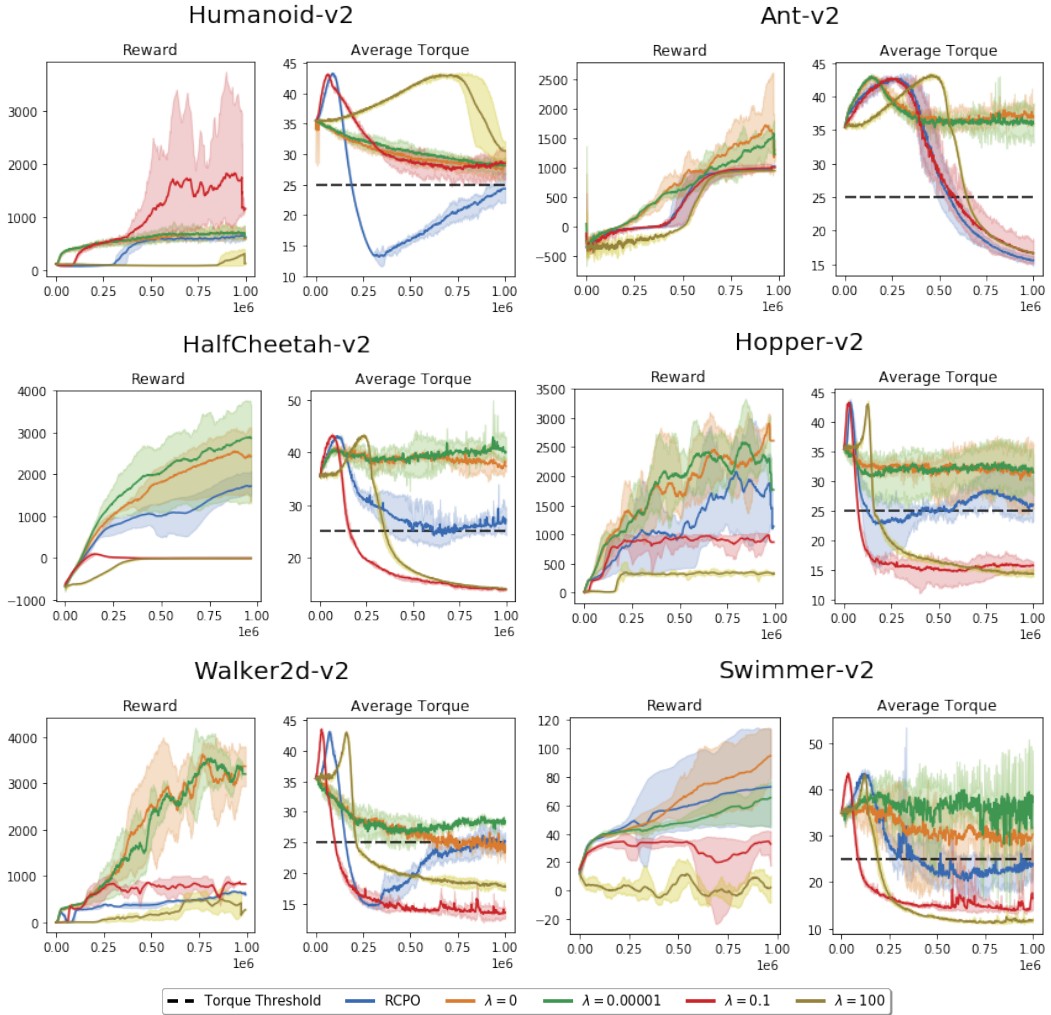

Figure 3: Mujoco with torque constraints. The dashed line represents the maximal allowed value. Results are considered valid only if they are at or below the threshold. RCPO is our approach, whereas each $\lambda$ value is a PPO simulation with a fixed penalty coefficient. Y axis is the average reward and the X axis represents the number of samples (steps).

2. Selecting a constant coefficient $\lambda$ such that the policy satisfies the constraint is not a trivial task, resulting in different results across domains (Achiam et al., 2017).

### 5.2.4 THE DRAWBACKS OF REWARD SHAPING

When performing reward shaping (selecting a fixed $\lambda$ value), the experiments show that in domains where the agent attains a high value, the penalty coefficient is required to be larger in order for the solution to satisfy the constraints. However, in domains where the agent attains a relatively low value, the same penalty coefficients can lead to drastically different behavior - often with severely sub-optimal solutions (e.g. Ant-v2 compared to Swimmer-v2).

Additionally, in RL, the value $(J_R^\pi)$ increases as training progresses, this suggests that a non-adaptive approach is prone to converge to sub-optimal solutions; when the penalty is large, it is plausible that at the beginning of training the agent will only focus on constraint satisfaction and ignore the underlying reward signal, quickly converging to a local minima.

## 6 Discussion

We introduced a novel constrained actor-critic approach, named 'Reward Constrained Policy Optimization' (RCPO). RCPO uses a multi-timescale approach; on the fast timescale an alternative, discounted, objective is estimated using a TD-critic; on the intermediate timescale the policy is learned using policy gradient methods; and on the slow timescale the penalty coefficient $\lambda$ is learned by ascending on the original constraint. We validate our approach using simulations on both grid-world and robotics domains and show that RCPO converges in a stable and sample efficient manner to a constraint satisfying policy.

An exciting extension of this work is the combination of RCPO with CPO (Achiam et al., 2017). As they consider the discounted penalty, our guiding signal, it might be possible to combine both approaches. Such an approach will be able to solve complex constraints while enjoying feasibility guarantees during training.

## 7 Acknowledgements

The authors would like to thank Nadav Merlis for the insightful discussions and helpful remarks during the writing process.

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

## A RCPO Algorithm

---
**Algorithm 2** RCPO Advantage Actor Critic
---
The original Advantage Actor Critic algorithm is in gray, whereas our additions are highlighted in black.

1: Input: penalty function $C(\cdot)$, threshold $\alpha$ and learning rates $\eta_1, \eta_2, \eta_3$
2: Initialize actor $\pi(\cdot|\cdot;\theta_p)$ and critic $V(\cdot;\theta_v)$ with random weights
3: Initialize $\lambda = 0$, $t = 0$, $s_0 \sim \mu$          ▷ Restart
4: **for** $T = 1, 2, ..., T_{max}$ **do**
5:      Reset gradients $d\theta_v \leftarrow 0$, $d\theta_p \leftarrow 0$ and $\forall i : d\lambda_i \leftarrow 0$
6:      $t_{start} = t$
7:      **while** $s_t$ not terminal and $t - t_{start} < t_{max}$ **do**
8:          Perform $a_t$ according to policy $\pi(a_t|s_t; \theta_p)$
9:          Receive $r_t$, $s_{t+1}$ and penalty score $\hat{C}_t$
10:          $t \leftarrow t + 1$
11:      $R = \begin{cases} 0 & , \text{ for terminal } s_t \\ V(s_t, \theta_v) & , \text{ otherwise} \end{cases}$
12:      **for** $\tau = t - 1, t - 2, ..., t_{start}$ **do**
13:          $R \leftarrow r_\tau - \lambda \cdot \hat{C}_\tau + \gamma R$          ▷ Equation 10
14:          $d\theta_p \leftarrow d\theta_p + \nabla_{\theta_p} \log \pi(a_\tau|s_\tau; \theta_p)(R - V(s_\tau; \theta_v))$
15:          $d\theta_v \leftarrow d\theta_v + \partial(R - V(s_\tau; \theta_v))^2 / \partial\theta_v$
16:      **if** $s_t$ is terminal state **then**
17:          $d\lambda \leftarrow -(C - \alpha)$          ▷ Equation 8
18:          $t \leftarrow 0$
19:          $s_0 \sim \mu$
20:      Update $\theta_v$, $\theta_p$ and $\lambda$
21:      Set $\lambda = \max(\lambda, 0)$          ▷ Ensure weights are non-negative (Equation 4)
---

## B Experiment details

### B.1 Mars Rover

The MDP was defined as follows:

$$r_{\text{step}} = -0.01, \ \ r_{\text{goal}} = 0, \ \ \delta = 0.05, \ \ \gamma = 0.99 \ .$$

In order to avoid the issue of exploration in this domain, we employ a linearly decaying random restart (Kakade and Langford, 2002). $\mu$, the initial state distribution, follows the following rule:

$$\mu = \begin{cases} uniform(s \in S) & \text{w.p. } \frac{1}{\#iteration} \\ s^* & else \end{cases}$$

where $S$ denotes all the non-terminal states in the state space and $s^*$ is the state at the top left corner (red in Figure 1). Initially the agent starts at a random state, effectively improving the exploration and reducing convergence time. As training progresses, with increasing probability, the agent starts at the top left corner, the state which we test against.

The A2C architecture is the standard non-recurrent architecture, where the actor and critic share the internal representation and only hold a separate final projection layer. The input is fully-observable, being the whole grid. The network is as follows:

| Layer | Actor | Critic |
|-------|-------|--------|
| 1 | CNN (input layers = 1, output layers = 16, kernel size = 5, stride = 3) | |
| 2 | CNN (input layers = 16, output layers = 32, kernel size = 3, stride = 2) | |
| 3 | CNN (input layers = 32, output layers = 32, kernel size = 2, stride = 1) | |
| 4 | Linear(input = 288, output = 64) | Linear(input = 288, output = 64) |
| 5 | Linear(input = 64, output = 4) | Linear(input = 64, output = 1) |
| LR | 1e-3 | 5e-4 |

between the layers we apply a ReLU non-linearity.

As performance is noisy on such risk-sensitive environments, we evaluated the agent every 5120 episodes for a length of 1024 episodes. To reduce the initial convergence time, we start $\lambda$ at 0.6 and use a learning rate $lr_\lambda = 0.000025$.

## B.2 ROBOTICS

For these experiments we used a PyTorch (Paszke et al., 2017) implementation of PPO (Kostrikov, 2018). Notice that as in each domain the state represents the location and velocity of each joint, the number of inputs differs between domains. The network is as follows:

| Layer | Actor | Critic |
|---|---|---|
| 1 | Linear(input = x, output = 64) | Linear(input = x, output = 64) |
| 2 | Linear(input = 64, output = 64) | Linear(input = 64, output = 64) |
| 3 | DiagGaussian(input = 64, output = y) | Linear(input = 64, output = 1) |
| LR | 3e-4 | 1.5e-4 |

where DiagGaussian is a multivariate Gaussian distribution layer which learns a mean (as a function of the previous layers output) and std, per each motor, from which the torque is sampled. Between each layer, a Tanh non-linearity is applied.

We report the online performance of the agent and run each test for a total of 1M samples. In these domains we start $\lambda$ at 0 and use a learning rate $lr_\lambda = 5e-7$ which decays at a rate of $\kappa = (1 - 1e - 9)$ in order to avoid oscillations.

The simulations were run using Generalized Advantage Estimation (Schulman et al., 2015b) with coefficient $\tau = 0.95$ and discount factor $\gamma = 0.99$.

## C    PROOF OF THEOREM 1

We provide a brief proof for clarity. We refer the reader to Chapter 6 of Borkar et al. (2008) for a full proof of convergence for two-timescale stochastic approximation processes.

Initially, we assume nothing regarding the structure of the constraint as such $\lambda_{\max}$ is given some finite value. The special case in which Assumption 2 holds is handled in Lemma 1.

The proof of convergence to a local saddle point of the Lagrangian (4) contains the following main steps:

1. **Convergence of $\theta$-recursion:** We utilize the fact that owing to projection, the $\theta$ parameter is stable. We show that the $\theta$-recursion tracks an ODE in the asymptotic limit, for any given value of $\lambda$ on the slowest timescale.

2. **Convergence of $\lambda$-recursion:** This step is similar to earlier analysis for constrained MDPs. In particular, we show that $\lambda$-recursion in (4) converges and the overall convergence of $(\theta_k, \lambda_k)$ is to a local saddle point $(\theta^*(\lambda^*, \lambda^*)$ of $L(\lambda, \theta)$.

**Step 1:** Due to the timescale separation, we can assume that the value of $\lambda$ (updated on the slower timescale) is constant. As such it is clear that the following ODE governs the evolution of $\theta$:

$$\dot{\theta}_t = \Gamma_\theta(\nabla_\theta L(\lambda, \theta_t)) \tag{12}$$

where $\Gamma_\theta$ is a projection operator which ensures that the evolution of the ODE stays within the compact and convex set $\Theta := \Pi_{i=1}^k \left[ \theta_{\min}^i, \theta_{\max}^i \right]$.

As $\lambda$ is considered constant, the process over $\theta$ is:

$$\theta_{k+1} = \Gamma_\theta[\theta_k + \eta_2(k)\nabla_\theta L(\lambda, \theta_k)]$$
$$= \Gamma_\theta[\theta_k + \eta_2(k)\nabla_\theta \mathbb{E}_{s\sim\mu}^{\pi_\theta} [\log \pi(s, a; \theta) [R(s) - \lambda \cdot C(s)]]]$$

Thus (6) can be seen as a discretization of the ODE (12). Finally, using the standard stochastic approximation arguments from Borkar et al. (2008) concludes step 1.

**Step 2:** We start by showing that the $\lambda$-recursion converges and then show that the whole process converges to a local saddle point of $L(\lambda, \theta)$.

The process governing the evolution of $\lambda$:

$$\lambda_{k+1} = \Gamma_\lambda[\lambda_k - \eta_1(k)\nabla_\lambda L(\lambda_k, \theta(\lambda_k))]$$
$$= \Gamma_\lambda[\lambda_k + \eta_1(k)(\mathbb{E}_{s\sim\mu}^{\pi_{\theta(\lambda_k)}}[C(s)] - \alpha)]$$

where $\theta(\lambda_k)$ is the limiting point of the $\theta$-recursion corresponding to $\lambda_k$, can be seen as the following ODE:

$$\dot{\lambda}_t = \Gamma_\lambda(\nabla_\lambda L(\lambda_t, \theta(\lambda_t))) \ . \tag{13}$$

As shown in Borkar et al. (2008) chapter 6, $(\lambda_n, \theta_n)$ converges to the internally chain transitive invariant sets of the ODE (13), $\dot{\theta}_t = 0$. Thus, $(\lambda_n, \theta_n) \to \{(\lambda(\theta), \theta) : \theta \in \mathbb{R}^k\}$ almost surely.

Finally, as seen in Theorem 2 of Chapter 2 of Borkar et al. (2008), $\theta_n \to \theta^*$ a.s. then $\lambda_n \to \lambda(\theta^*)$ a.s. which completes the proof.

## D    Proof of Lemma 1

The proof is obtained by a simple extension to that of Theorem 1. Assumption 2 states that any local minima $\pi_\theta$ of 2 satisfies the constraints, e.g. $J_C^{\pi_\theta} \le \alpha$; additionally, Lee et al. (2017) show that first order methods such as gradient descent, converge almost surely to a local minima (avoiding saddle points and local maxima). Hence for $\lambda_{\max} = \infty$ (unbounded Lagrange multiplier), the process converges to a fixed point $(\theta^*(\lambda^*), \lambda^*)$ which is a feasible solution.

## E    Proof of Theorem 2

As opposed to Theorem 1, in this case we are considering a three-timescale stochastic approximation scheme (the previous Theorem considered two-timescales). The proof is similar in essence to that of Prashanth and Ghavamzadeh (2016).

The full process is described as follows:

$$\lambda_{k+1} = \Gamma_\lambda[\lambda_k + \eta_1(k)(\mathbb{E}_{s\sim\mu}^{\pi_{\theta(\lambda_k)}}[C(s)] - \alpha)]$$

$$\theta_{k+1} = \Gamma_\theta[\theta_k + \eta_2(k)\nabla_\theta \mathbb{E}_{s\sim\mu}^{\pi_\theta}\left[\log \pi(s,a;\theta)\hat{V}(\lambda, s_t; v_k)\right]]$$

$$v_{k+1} = v_k - \eta_3(k)\left[\partial(\hat{r} + \gamma\hat{V}(\lambda, s'; v_k) - \hat{V}(\lambda, s; v_k))^2/\partial v_k\right]$$

**Step 1:** The value $v_k$ runs on the fastest timescale, hence it observes $\theta$ and $\lambda$ as static. As the TD operator is a contraction we conclude that $v_k \to v(\lambda, \theta)$.

**Step 2:** For the policy recursion $\theta_k$, due to the timescale differences, we can assume that the critic $v$ has converged and that $\lambda$ is static. Thus as seen in the proof of Theorem 1, $\theta_k$ converges to the fixed point $\theta(\lambda, v)$.

**Step 3:** As shown previously (and in Prashanth and Ghavamzadeh (2016)), $(\lambda_n, \theta_n, v_n) \to (\lambda(\theta^*), \theta^*, v(\theta^*))$ a.s.

Denoting by $\Theta = \{\theta : J_C^{\pi_\theta} \le \alpha\}$ the set of feasible solutions and the set of local-minimas of $J_{C_\gamma}^{\pi_\theta}$ as $\Theta_\gamma$. We recall the assumption stated in Theorem 2:

**Assumption 4.** $\Theta_\gamma \subseteq \Theta$.

Given that the assumption above holds, we may conclude that for $\lambda_{\max} \to \infty$, the set of stationary points of the process are limited to a sub-set of feasible solutions of (4). As such the process converges a.s. to a feasible solution.

We finish by providing intuition regarding the behavior in case the assumptions do not hold.

1. **Assumption 2 does not hold:** As gradient descent algorithms descend until reaching a (local) stationary point. In such a scenario, the algorithm is only ensured to converge to some stationary solution, yet said solution is not necessarily a feasible one.

   As such we can only treat the constraint as a regularizing term for the policy in which $\lambda_{\max}$ defines the maximal regularization allowed.

2. **Assumption 4 does not hold:** In this case, it is not safe to assume that the gradient of (2) may be used as a guide for solving (3). A Monte-Carlo approach may be used (as seen in Section 5.1) to approximate the gradients, however this does not enjoy the benefits of reduced variance and smaller samples (due to the lack of a critic).

