# OpenReview forum: "Reward Constrained Policy Optimization"
_ICLR.cc/2019/Conference_

### Official Review · AnonReviewer3 · 2018-10-28
**Important topic but limited experimental validation**

**Rating:** 6
**Confidence:** 2

**Review:**

The paper introduces RCPO, a model-free deep RL algorithm for learning optimal policies that satisfy some per-state constraint on expectation. The derivation of the algorithm is quite straightforward, starts from the definition of constrained optimization problem, and proceed by forming and optimizing the Lagrangian. Additionally, a value function for the constraint is learned. The algorithm is only compared to a baseline optimizing the Lagrangian directly using Monte-Carlo sampling.

The paper has two major problems. First, while the derivation of the method makes intuitively sense, it is supported by vaguely stated theorems, which mixes rigorous guarantees with practical approximations. For example, Equation 4 assumes strong duality. How would the result change if weak duality was used instead? The main result in Theorem 1 makes the assumption that dual variable is constant with respect the policy, which might be true in practice, but it is not obvious how the approximation affects the theory. Further, instead of simply referring to prior convergence results, I would strongly suggest including the exact prior theorems and assumptions in the appendix.

The second problem is the empirical validation, which is incomplete and misleading. Constrained policy optimization is not a new topic (e.g. work by Achiam et al.), so it is important to compare to the prior works. It is stated in the paper that the prior methods cannot be used to handle mean value constraints. However, it would be important to include experiments that can be solved with prior methods too, for example the experiments in Achiam at al. for proper comparison. The results in Table 2 are confusing: what makes the bolded results better than the others? If the criterion is highest return and torque < 25%, then \lambda=0.1 should be chosen for Hopper-v2. Also, The results seem to have high variance, and judging based on Table 2 and Figure 3, it is not obvious how well RCPO actually works.

To summarize, while the topic is undoubtedly important, the paper would need be improved in terms of better differentiating the theory from practice, and by including a rigorous comparison to prior work.

Minor points:
- What is exactly the difference between discounted sum constraint and mean value constraint?
- Could consider use colors in Table 1.
- Section 4.1.: What does “... enables training using a finite number of samples” exactly mean in this case?
- Table 2: The grid for \lambda is too sparse.
- Proof of Theorem 1: What does it mean \theta to be stable?
- Proof of Theorem 2: “Theorem C” -> “Theorem 1”

---

> ### Author Response · Authors · 2018-11-13
> **Thank you for your helpful comment, we address your concerns in our comment below**
>
> We thank the reviewer for his/her helpful comments and feedback.
>
> Lagrange relaxation:
> It is important to differentiate between our derivation and the guarantees in the case where strong duality holds. Our derivation does not assume strong duality (in fact, we have weak duality as we are assuming a non-convex RL setting). Hence, our algorithm and (4) do not result in the same solution (by definition of the Lagrange relaxation) [3]. Our algorithm ensures convergence to a locally-optimal feasible solution - which tends to be a reasonable, constraint satisfying solution as shown in our experiments.
> Theorem 1: This theorem is an extension to the well known two-timescale stochastic approximation theorem from Borkar [1]. It states that due to the timescale separation between the dual-variable signal and the policy, the dual-variable is quasi-static w.r.t. the policy. It does not imply that the dual variable is constant, as it is also being iteratively updated. Rather, since it is on the slower timescale, the policy observes it as changing “slowly” enabling it to be seen as being quasi-static.
>
> Empirical validation:
> The experiments aim to show the benefit of our approach to solving general constraints using a penalty signal. For this reason, we compare to two common approaches which are able to do so (a simple adaptation of Proximal Policy Optimization (PPO) which we refer to as reward shaping, and the standard Lagrange dual optimization). We, therefore, do compare our work to relevant prior methods. Achaim’s work is fundamentally different and therefore would be an "apples to oranges" comparison.
>
> Results: The point we are making in Table 2 is that, without any manually chosen grid search over lambdas, RCPO can attain competitive performance compared to the optimal lambda across all domains when taking both (1) constraint satisfaction and (2) average reward into account. On a subsequent inspection, we agree that it is unclear for specifically Hopper v2 which method should be bolded (and will be noted in the final version), but we do want to emphasize that the point of this Table is to show the flexibility of our technique to multiple domains without having to go through the cumbersome process of hand-tuning the dual parameters (i.e., reward shaping).
>
> Variance: The variance is an algorithmic artifact which is also apparent in the reward shaping variants (constant \lambda) as well as the original PPO work [4]. There are techniques that we may be able to employ to reduce the variance, but that is for future work.
>
> Minor remarks:
> - Mean vs Discounted constraints: A mean valued constraint is equal to (1 / N) sum_{i=1}^N C_i while a discounted sum is equal to sum_{i=1}^N \gamma^i C_i. While an average value is intuitive to define and understand (e.g. the average torque the agent applies to the motors), the discounted sum is not. A simple example is that in very long trajectories, you become agnostic as to how the agent behaves as the discount factor to the power of the trajectory length (\gamma^t) goes to 0 very fast.
> - Finite samples: Without a critic, the policy gradient algorithm is required to sample an entire trajectory [2], e.g. monte carlo sampling. A critic enables you to bootstrap the estimated future value, this enables you to train on \sum_{i=1}^N \gamma^i r_i + \gamma^{N+1} V(s’) instead of \sum_{i=1}^\infty \gamma^i r_i.
> - Sparse lambda: The aim of the experiments was to show that it is not clear how to select an appropriate constant lambda (penalty) value. Once a penalty value is found, which works for one task, it does not necessarily transfer to subsequent tasks. This is in contrast to RCPO that learns the lambda that works for each task.
> - Stability of theta: Stability of \theta is in the fact that it is projected into a closed and compact set. The iterates do not diverge. In practice, an algorithm like PPO keeps the iterates stable as they are kept within a ‘trust region’.
>
> [1] V. Borkar - Stochastic approximation
> [2] Ronald J Williams - Simple statistical gradient-following algorithms for connectionist reinforcement learning
> [3] D Bertesekas - Nonlinear programming
> [4] John Schulman, Filip Wolski, Prafulla Dhariwal, Alec Radford, and Oleg Klimov - Proximal policy optimization algorithms

---

> > ### Comment · AnonReviewer3 · 2018-11-22
> > **Thank you for the clarifications**
> >
> > The response clarifies most of my concerns. I have adjusted the rating accordingly.

---

### Official Review · AnonReviewer2 · 2018-11-03
**Well written, addresses relevant problem in reinforcement learning, proposes interesting solution**

**Rating:** 7
**Confidence:** 2

**Review:**

This work tackles the difficult problem of solving Constrained Markov Decision Processes. It proposes the RCPO algorithm as a way to solving CMDP. The benefits of RCPO is that it can handle general constraints, it is reward agnostic and doesn't require prior knowledge. The key is that RCPO trains the actor and critic using an alternative penalty guiding signal.

Pros:
- the motivations for the work are clearly explained and highly relevant
- comprehensive overview of related work
- clear and consistent structure and notations throughout
- convergence proof is provided under certain assumptions


Cons:
- no intuition is given as to why RCPO isn't able to outperform reward shaping in the Walker2d-v2 domain

minor remarks:
- in Table 2, it would be good if the torque threshold value appeared somewhere
- in Figure 3, the variable of the x-axis should appear either in the plots or in the legend
- in appendix B, r_s should be r_step and r_T should be r_goal to stay consistent with notation in 5.1.1

---

> ### Author Response · Authors · 2018-11-13
> **Thank you for your helpful review**
>
> We thank the reviewer for his/her helpful comments and feedback.
>
> The intuition behind the inability of RCPO to outperform the reward shaping agent in Walker2d is that the unconstrained algorithm (\lambda = 0) results in a feasible solution. This means that for the default MDP without any external intervention, the resulting optimal policy is feasible.
>
> Thank you for the additional minor points. We will incorporate these changes into the paper.

---

### Official Review · AnonReviewer1 · 2018-11-06
**Interesting paper, with some details not so clear to me**

**Rating:** 6
**Confidence:** 4

**Review:**

This paper proposed a general framework for policy optimization of constrained MDP. Compared with the traditional methods, such as Lagrangian multiplier methods and trust region approach, this method shows better empirical results and theoretical merits.

Major concerns:
My major concern is about the time-scales. RCPO algorithm, by essence, is multi-scale, which usually has a stringent requirement on the stepsizes and is difficult to tune in practice to obtain the optimal performance. The reviewer would like to see how robust the algorithm is if the multi-time-scale condition is violated, aka, is the algorithm's performance robust to the stepsizes?

My second concern is the algorithm claims to be the first one to handle mean-value constraint without reward shaping. I did not get the reason why (Dalal et al. 2018) and (Achiam et al., 2017) cannot handle this case. Can the authors explain the reason more clearly?

Some minor points:
The experiments are somewhat weak. The author is suggested to compare with more baseline methods. Mujoco domain is not a very difficult domain in general, and the authors are suggested to compare the performance on some other benchmark domains.

This paper needs to consider the cases where the constraints are the squares of returns, such as the variance. In that case, computing the solution often involves double sampling problem. Double sampling problem is usually solved by adding an extra variable at an extra time scale (if gradient or semi-gradient methods are applied), such as in many risk-sensitive papers.​

---

> ### Author Response · Authors · 2018-11-13
> **Thank you for your helpful review, clarification provided in the comment**
>
> We thank the reviewer for his/her helpful comments and feedback.
>
> Multiple timescales:
> As the reviewer correctly pointed out, there are certain requirements placed on the step-sizes. It is important to note though, that these requirements are standard assumptions that are used in numerous works [e.g.,1, 2] to ensure convergence of the algorithm. While tuning the step-sizes requires hyperparameter sweeps, it is common practice to do so. It may also be possible to remove one of the timescales by using techniques such as population-based training or parameter exploring policy gradients [3] (both of which we are looking into). However, the focus of our work is a proof-of-concept algorithm to show the ability of our agent to incorporate constraints directly into the reward function and learn the Lagrange parameters on a separate timescale. While it is desirable to reduce the number of timescales, empirically we observed good performance and we intend to look into this direction as future work.
>
> Mean-valued constraints:
> As we alluded to in our comparison table, algorithms [4] and [5] are incapable of handling mean-valued constraints. [4] considers the discounted cost-to-go in order for the TRPO algorithm to ensure a safe update; and [5] performs a one-step lookahead in an attempt to ensure constraint satisfaction of critical constraints (for instance there is a region the agent is not allowed to enter).
>
> Experiments:
> The Mujoco platform is considered to be a standard and challenging benchmark in continuous control problems (including constraint-based papers [4,5]). Much research coming out of the control and reinforcement learning community compare their algorithms on these Mujoco baseline domains. This is the reason why we chose these domains.
>
> Square valued constraints: Such constraints, e.g., variance, are indeed a promising research avenue. However, as you correctly mentioned, they require special attention (an extra variable, and potentially an additional timescale, to ensure an unbiased estimate). We focused specifically on mean-value and probability constraints. We should be able to adapt our approach to handling squared value constraints, and this is definitely a problem we intend to look into.
>
> [1] Shalabh Bhatnagar and K. Lakshmanan - An Online Actor–Critic Algorithm with Function Approximation for Constrained Markov Decision Processes
> [2] V. Borkar - An actor-critic algorithm for constrained Markov decision processes
> [3] Senkhe et. al., Parameter Exploring Policy Gradients
> [4] Joshua Achiam, David Held, Aviv Tamar, and Pieter Abbeel. Constrained policy optimization
> [5] Gal  Dalal,  Krishnamurthy  Dvijotham,  Matej  Vecerik,  Todd  Hester,  Cosmin  Paduraru, and Yuval Tassa.Safe  exploration  in  continuous  action  spaces

---

### Public Comment · (anonymous) · 2018-11-05
**Interesting paper, with some details not so clear to me**

This paper proposed a general framework for policy optimization of constrained MDP. Compared with the traditional methods, such as Lagrangian multiplier methods and trust region approach, this method shows better empirical results and theoretical merits.


Major concerns:

My major concern is about the time-scales. RCPO algorithm, by essence, is multi-scale, which usually has a stringent requirement on the stepsizes and is difficult to tune in practice to obtain the optimal performance. The reviewer would like to see how robust the algorithm is if the multi-time-scale condition is violated, aka, is the algorithm's performance robust to the stepsizes?

My second concern is the algorithm claims to be the first one to handle mean-value constraint without reward shaping. I did not get the reason why (Dalal et al. 2018) and (Achiam et al., 2017) cannot handle this case. Can the authors explain the reason more clearly?

Some minor points:

The experiments are somewhat weak. The author is suggested to compare with more baseline methods. Mujoco domain is not a very difficult domain in general, and the authors are suggested to compare the performance on some other benchmark domains.

This paper needs to consider the cases where the constraints are the squares of returns, such as the variance. In that case, computing the solution often involves double sampling problem. Double sampling problem is usually solved by adding an extra variable at an extra time scale (if gradient or semi-gradient methods are applied), such as in many risk-sensitive papers.

---

### Meta-Review · Area_Chair1 · 2018-12-10
**Novel approach to constrained policy optimization with room for improvement in understanding the strategy**

**Confidence:** 4
**Recommendation:** Accept (Poster)

**Metareview:**

This work is novel, reasonably clearly written with a thorough literature survey. The proposed approach also empirically seems promising. The paper could be improved with a bit more discussion about the sensitivity, particularly as a two-timescale approach can be more difficult to tune.